# A single cysteine residue in vimentin regulates long non-coding RNA *XIST* to suppress epithelial–mesenchymal transition and stemness in breast cancer

Saima Usman[1], William Andrew Yeudall[2,3], Muy-Teck Teh[1], Fatemah Ghloum[1], Hemanth Tummala[4], Ahmad Waseem[1]*

[1]Centre for Oral Immunobiology and Regenerative Medicine, Institute of Dentistry, Barts and The London School of Medicine and Dentistry, Queen Mary University of London, London, United Kingdom; [2]Department of Oral Biology and Diagnostic Sciences, The Dental College of Georgia, Augusta University, Augusta, United States; [3]Georgia Cancer Center, Augusta University, Augusta, United States; [4]Centre for Genomics and Child Health, Blizard Institute, Barts and The London School of Medicine and Dentistry, Queen Mary University of London, London, United Kingdom

## eLife Assessment

This **valuable** study reveals that the structural protein vimentin promotes the epithelial–mesenchymal transition in breast cancer cells. Utilising robust and validated methodologies, the data collected provide a **solid** foundation for further investigation into metastasis models. This work will be of significant interest to researchers in the field of breast cancer.

*For correspondence:
a.waseem@qmul.ac.uk

**Abstract** Vimentin is a type III intermediate filament (IF) protein that is induced in a large number of solid tumours. A single cysteine at position 328 in vimentin plays a crucial role in assembly, organisation, and stability of IFs. However, its exact function during epithelial–mesenchymal transition (EMT) and cancer progression has not been investigated. To investigate this, we have transduced wildtype (WT) and C328S vimentin separately in MCF-7 cells that lack endogenous vimentin. The expression of C328-VIM impacted vimentin–actin interactions and induced EMT-like features that include enhanced cell proliferation, migration, and invasion accompanied by reduced cell adhesion when compared to the wildtype cells. Functional transcriptomic studies confirmed the upregulation of EMT and mesenchymal markers, downregulation of epithelial markers, as well as acquisition of signatures associated with cancer stemness (*CD56, POU5F1, PROCR,* and *CD49f*), thus transforming MCF-7 cells from oestrogen-positive to triple-reduced (*ESR1, PGR,* and *HER2*) status. We also observed a stark increase in the expression of long non-coding RNA, *XIST,* in MCF-7 cells expressing C328-VIM. Targeting the mutant vimentin or *XIST* by RNA interference partially reversed the phenotypes in C328-VIM-expressing MCF-7 cells. Furthermore, the introduction of C328-VIM cells into nude mice promoted tumour growth by increasing cancer stemness in an oestrogen-independent manner. Altogether, our studies provide insight into how cysteine 328 in vimentin dictates mechano-transduction signals to remodel actin cytoskeleton and protect against EMT and cancer growth via modulating lncRNA *XIST*. Therefore, targeting vimentin and/or *XIST* via RNA interference should be a promising therapeutic strategy for breast cancer treatment.

## Introduction

Vimentin is a type III intermediate filament (IF) protein normally expressed in mesenchymal cells. Due to its diverse pathophysiological role(s), it is one of the most extensively studied IFs. It has been proposed to play a role in epithelial-to-mesenchymal transition (EMT) and cancer metastasis (*Usman et al., 2021*). It is a structurally dynamic protein active in cell mechanics (*Patteson et al., 2020*), normal positioning of cell organelles, nuclear and DNA integrity (*Danielsson et al., 2018*), stress response (*Mónico et al., 2019*), autophagy (*Surolia and Antony, 2022*), cell proliferation, migration (*Wang et al., 2019*), adhesion, invasion (*Wang et al., 2021*), signalling, angiogenesis (*Chen et al., 2021*), and immune responses (*Ridge et al., 2022*). The complete crystal structure of vimentin is still not available; however, crystal structures of small segments have been investigated (*Strelkov et al., 2002*). Structurally, vimentin monomer has a central α-helical rod domain flanked by non-helical head and tail domains on each side (*Chernyatina et al., 2012*). Studies have described its polymerisation in which monomers assemble in parallel to form dimers, with two dimers assembling in antiparallel to form tetramers (*Herrmann and Aebi, 2016*). Eight tetramers combine to form unit length filaments (ULFs) that further assemble head to tail into 10 nm compact mature filaments (*Nunes Vicente et al., 2022*). IFs lack polarity and in mature filaments subunits continue to assemble at both ends, and they can be exchanged anywhere along the whole filament length, a process that requires ATP (*Robert et al., 2015*). Vimentin directly interacts with actin through its tail domain to maintain mechanical integrity of the cell and cytoskeletal crosstalk (*Esue et al., 2006*).

Vimentin has a single cysteine residue at position 328 in the rod domain, which is reported to be a stress sensor induced by oxidants and electrophiles, resulting in its zinc-mediated modifications and consequently disassembly or rearrangement of filaments as a stress response (*Mónico et al., 2019*; *Pérez-Sala et al., 2015*). Different modifications of C328 that have been reported during cellular senescence, rheumatoid arthritis, cataract, and atherosclerosis (*Viedma-Poyatos et al., 2020*). Any mutation or modification in C328 can hamper the intracellular stress response of vimentin. As a result, multiple cellular functions can be disrupted due to the inability of the cell to sense and respond to stress. Therefore, this residue has been reported to have widespread pathophysiological implications and is considered a flash point for stress ignition (*Viedma-Poyatos et al., 2020*). Although the role

**Table 1.** Summary of cell lines used in this study.

| Name of cell line | Insert | Drug selection | Retrovirus vector | Parent cell line |
|---|---|---|---|---|
| MCF-7CV | None; vector | Hygromycin | pLPChygro | MCF-7 |
| MCF-7WT | WT-VIM | Hygromycin | pLPChygro-VIM | MCF-7 |
| MCF-7C328S | C328S-VIM | Hygromycin | pLPChygro-C328S-VIM | MCF-7 |
| MCF-7Y117L | Y117L-VIM | Hygromycin | pLPChygro-Y117L-VIM | MCF-7 |
| MCF-7DMT (double mutant) | Y117L,C328S-VIM | Hygromycin | pLPChygro-DMT-VIM | MCF-7 |
| MCF-7C328S_shNTC | Non-target control oligos | Puromycin | pSiren-Retro-Q vector | MCF-7C328S |
| MCF-7C328S_shVIM | Vimentin specific short hairpin oligos | Puromycin | pSuper.retro.puro | MCF-7 C328S |
| MCF-7C328S_shNTC | Non-targeting control oligos | Puromycin | pSuper.retro.puro | MCF-7 C328S |
| MCF-7C328S_shXIST-1 | *XIST* specific short hairpin 1 | Puromycin | pSuper.retro.puro | MCF-7 C328S |
| MCF-7C328S_shXIST-2 | *XIST* specific short hairpin 2 | Puromycin | pSuper.retro.puro | MCF-7 C328S |
| MCF-7C328S_shXIST-3 | *XIST* specific short hairpin 3 | Puromycin | pSuper.retro.puro | MCF-7 C328S |
| MCF-7C328S_shXIST-4 | *XIST* specific short hairpin 4 | Puromycin | pSuper.retro.puro | MCF-7 C328S |
| MCF-7WT+AcGFP-C328S-VIM | C328S-VIM | Puromycin | pLPCpuro-AcGFP-GS10 | MCF-7WT |
| MCF-7WT+C328S-VIM | C328S-VIM | Puromycin | pLPCpuro | MCF-7WT |
| HFF-AcGFP-C328S-VIM | C328S-VIM | Puromycin | pLPCpuro-AcGFP-GS10 | HFF-1 |
| HFF-C328S-VIM | C328S-VIM | Puromycin | pLPCpuro | HFF-1 |
| A431-C328S-VIM | C328S-VIM | Puromycin | pLPCpuro | A431 |

of vimenetin C328 in response to cellular stress has been studied in some detail, its role in regulating EMT, tumour growth, and progression has not been studied.

To investigate the functional implications of vimentin residue C328 in regulating EMT, cancer progression, and stemness, we expressed C328S vimentin in MCF-7 cells, a vimentin-deficient cell line (*Usman et al., 2022a*) commonly employed to study EMT (*Guttilla et al., 2012*; *Kondaveeti et al., 2015*). Our findings reveal that C328S-VIM altered cell morphology with disorganised F-actin and induced EMT and cancer stem cell characteristics. A highly significant observation was the upregulation of long non-coding RNA (lncRNA), *XIST,* in C328S-VIM-expressing MCF-7 cells, which are normally oestrogen-dependent for tumorigenesis (*Soule and McGrath, 1980*), becoming oestrogen-independent by C328S-VIM in nude mice. We show that C328S-VIM can trigger an EMT programme and enhance stemness in MCF-7 cells, in part via *XIST* upregulation. These findings have far-reaching implications, particularly considering the numerous vimentin variants reported in the vicinity of the C328 residue in several solid tumours (*Usman et al., 2021*).

## Results

### C328S mutant vimentin affects interaction with actin in silico and F-actin formation in cells

We selected MCF-7, a simple epithelial breast carcinoma cell line, which lacks endogenous vimentin (*Usman et al., 2022a*), and transduced it with the full-length vimentin (wildtype, WT-VIM/pLPChygro-VIM) and the mutant vimentin (mutant, C328S-VIM/pLPChygro-C328S-VIM) retroviruses (*Table 1*). The expression of WT-VIM and C328S-VIM in MCF-7 cells was confirmed by qPCR (*Figure 1A*) and western blotting (*Figure 1B*). The quantification of the relevant bands showed that the wildtype and mutant vimentin proteins were equally expressed in MCF-7 cells (*Figure 1C*). Immunostaining of the wildtype vimentin showed a fully extended vimentin network from perinuclear to peripheral cell boundaries, whereas in cells transduced with the C328S mutant, the vimentin was more condensed around the perinuclear area (*Figure 1D*).

To confirm that C328S mutant vimentin was able to form filaments, A431 cells, which are devoid of endogenous vimentin (*Usman et al., 2022a*), were transduced with C328S-VIM. The immunostaining with mouse anti-vimentin V9 (referred to as V9 from now on) followed by AF-488-labelled goat anti-mouse secondary antibody showed normal appearing polymerised IFs in A431 cells (*Figure 1—figure supplement 1A, compare a–c with d–f*). This shows that C328S-VIM retains its ability to polymerise into filaments both in A431 and MCF-7 cells; however, in MCF-7 these filaments were drastically reorganised, affecting the cell shape (*Figure 1D*). Furthermore, to confirm that C328S-VIM does not disrupt pre-existing vimentin filaments (dominant negative), we transduced HFF-1 and MCF-7 cells expressing WT vimentin with either untagged C328S-VIM (*Figure 1—figure supplement 1B, g–l*) or AcGFP fused with C328S-VIM at its N-terminus (AcGFP-C328S-VIM) (*Figure 1—figure supplement 1C, m–r*). Cells were immunostained with V9 antibody. Our data showed that both untagged and AcGFP-C328S-VIM constructs were not dominant negative as they did not disturb the pre-existing filaments and appeared to integrate into the pre-existing network. However, the limitation of using the V9 antibody was that it detected both endogenous vimentin and C328S-VIM.

Next, to assess the impact of the C328S mutation on vimentin's interaction with actin, in silico structural analysis was conducted for both the WT and mutant vimentin proteins. The Protein Data Bank (PDB) file corresponding to the wildtype vimentin (PDB ID: P08670) was obtained from the RCSB PDB database (https://www.rcsb.org/). The specific mutation of interest, C328S, which replaces the cysteine residue at position 328 with a serine, was introduced into the WT structure, and energy minimisation steps were performed for best rotor fit to optimise the geometry of the mutant protein and resolve any steric clashes or unfavourable conformations resulting from the mutation. Subsequent structural simulation analyses were performed to evaluate the interactions between vimentin (both WT and C328S) and actin. These analyses focused on key binding sites, interaction energy, and potential structural changes caused by the C328S substitution. It should be noted that the thiol group of C328 enables specific hydrogen bonding and potential disulphide-mediated interactions with actin (*González-Jiménez et al., 2023*). The computational results revealed that the C328S mutation induced alterations within the coiled-coil rod domain of vimentin, particularly in regions critical for actin binding. The substitution of cysteine with serine likely altered the conformation of hydrogen

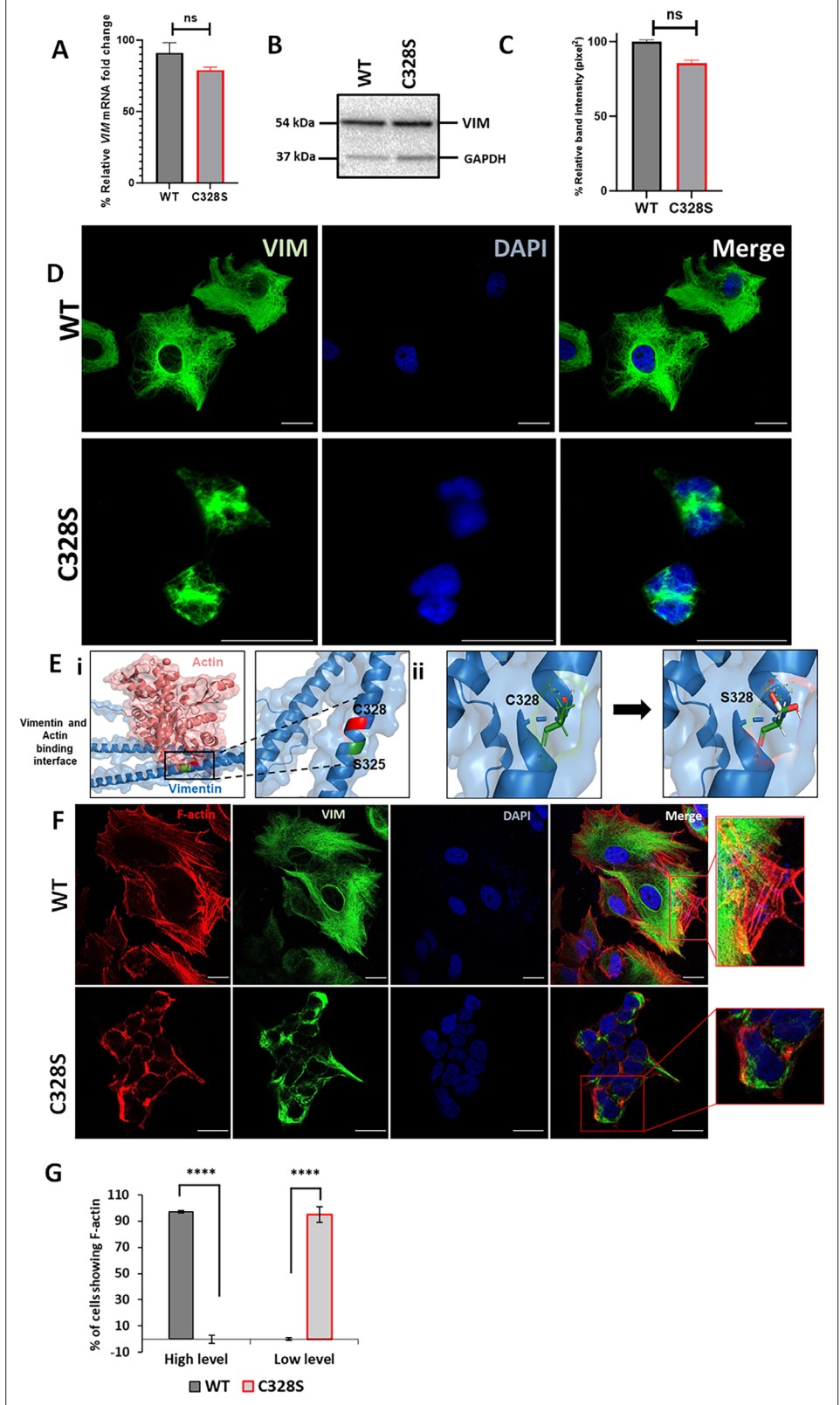

**Figure 1.** C328S mutant vimentin affects interaction with actin in silico and F-actin formation in cells. (**A**) Relative fold change of *VIM* mRNA in WT-VIM and C328S-VIM-expressing MCF-7 cells normalised to *POLR2A* and *YAP1*. (**B**) Expression of WT-VIM and C328S-VIM in MCF-7 cells by western blotting (original blots in *Figure 1—source data 1* and *Figure 1—source data 2*). (**C**) Quantification of vimentin using ImageJ. (**D**) Immunostaining of MCF-7

*Figure 1 continued on next page*

*Figure 1 continued*

cells expressing WT-VIM and C328S-VIM. Cells were immunostained with mouse anti-vimentin V9, AF-488-labelled goat anti-mouse showing green fluorescence. Nuclei were stained with DAPI in blue, and the overlapping images are shown as Merge. Leica DM4000B Epi-fluorescence microscope was used for imaging (scale bar = 20 µm; the scale bar in MCF-7C328S cells is much longer as these cells were reduced in size). (**E**) In silico modelling of actin binding at the interface where C328 and S325 residues are located. The solid rectangular area in panel (**i**) is zoomed in the accompanying panel. The red colour indicates the C328 residue site, whereas the green colour indicates the S325 residue site. (**ii**) In silico modelling of C328 and S328 residue sites and rotamer conformations further zoomed in from the panel (**i**). The green colour indicates hydrogen bonds, the white colour indicates covalent bonds, and the red colour indicates oxygen atom. (**F**) Immunostaining of MCF-7 cells expressing WT- and C328S-VIM. Cells were immunostained with AF568 Phalloidin (red colour) and anti-vimentin (green colour) antibody. Nuclei were in blue, and the overlapping images are shown as Merge. Images were taken using the Zeiss 880 laser scanning confocal microscope with Fast Airyscan and Multiphoton (inverted) system (scale bar = 20 µm). (**G**) Percentage of cells expressing F-actin in WT and C328 cells (see *Figure 1—figure supplement 2* for clarification of high and low expression). Student's *t*-test was used to calculate p values using Microsoft Excel and are given by asterisks (****$p<0.0001$). Statistical analyses: $n = 3$, error bars = ± SEM, ns = not significant, number of cells counted = 200.

The online version of this article includes the following source data and figure supplement(s) for figure 1:

**Source data 1.** Full-size western blot indicating the relevant bands cropped for *Figure 1B*.

**Source data 2.** Original file for western blot analysis displayed in *Figure 1B*.

**Figure supplement 1.** Immunostaining of vimentin filaments.

**Figure supplement 2.** Actin filaments in cells containing wildtype and C328S vimentin.

bonds and hydrophobic contacts, thus weakening the overall binding affinity of the mutant vimentin for actin (*Figure 1E*).

To further test this, we investigated C328S mutation-mediated structural changes in the cytoskeleton by staining these cells with Phalloidin that selectively labels F-actin. Quantification of stress fibre/F-actin staining pattern as described previously (*Avraham-Chakim et al., 2013*) confirmed that actin morphology and organisation was altered with low-level F-actin/stress fibre staining with aggregates/fragments at the cortical margins of the cells. No stress fibres were observed in the centres of the cells in C328S-VIM-expressing cells compared with high-level F-actin/stress fibre staining in WT (*Figure 1F and G*, *Figure 1—figure supplement 2; compare A with B*). These data show that C328S-VIM when expressed in MCF-7 cells alters cell morphology through remodelling of the actin cytoskeleton.

## C328S-VIM impacts cell adhesion, proliferation, and migration by altering MCF-7 cell morphology

To compare the effect of C328S-VIM on the MCF-7 cell morphology (*Figure 2A*), cells were fixed and stained with CellMask Deep Red dye and counterstained nuclei with DAPI (*Figure 2B*). Different cell morphological features, such as nuclear perimeter, cell diameter, nucleus/cell area, cell major axis, cell major axis angle, and cell minor axis, were analysed. C328S-VIM-expressing MCF-7 cells showed more cellular projections when compared to WT and analysis of nuclear area, nuclear form factor, nuclear major axis, nuclear minor axis, cell area, cell diameter, cell perimeter, cell minor axis, cell major axis, and cell form factor were significantly ($p<0.05$) increased (*Figure 2C*), while the nucleus/cell area ratio and cell compactness were significantly decreased in cells expressing C328S-VIM in comparison to WT-VIM-expressing cells (*Figure 2D*). These results indicate that the expression of C328S-VIM significantly altered nuclear and cell perimeter as evident by more ruffled margins in MCF-7 cells. Next, we investigated the rate of proliferation between WT and C328S cells by colony formation (*Figure 2E and F*), MTT (*Figure 2G*), and CyQUANT proliferation assays (*Figure 2H*). The analyses showed a highly significant increase in cell proliferation and mitochondrial activity in C328S compared to WT cells. Furthermore, C328S-VIM-expressing MCF-7 cells showed reduced adhesion without substrate coating on the culture vessel as determined by the CyQUANT cell adhesion assay (*Figure 2I*). However, when the tissue culture plates were coated with different substrates such as laminin, fibronectin, and collagen, there was no significant difference in cell adhesion (*Figure 2J*). The chemotactic migration of the cells towards foetal calf serum (FCS) through 8 µm pore size culture inserts showed the number of cells migrating through the membrane was significantly higher ($p<0.05$)

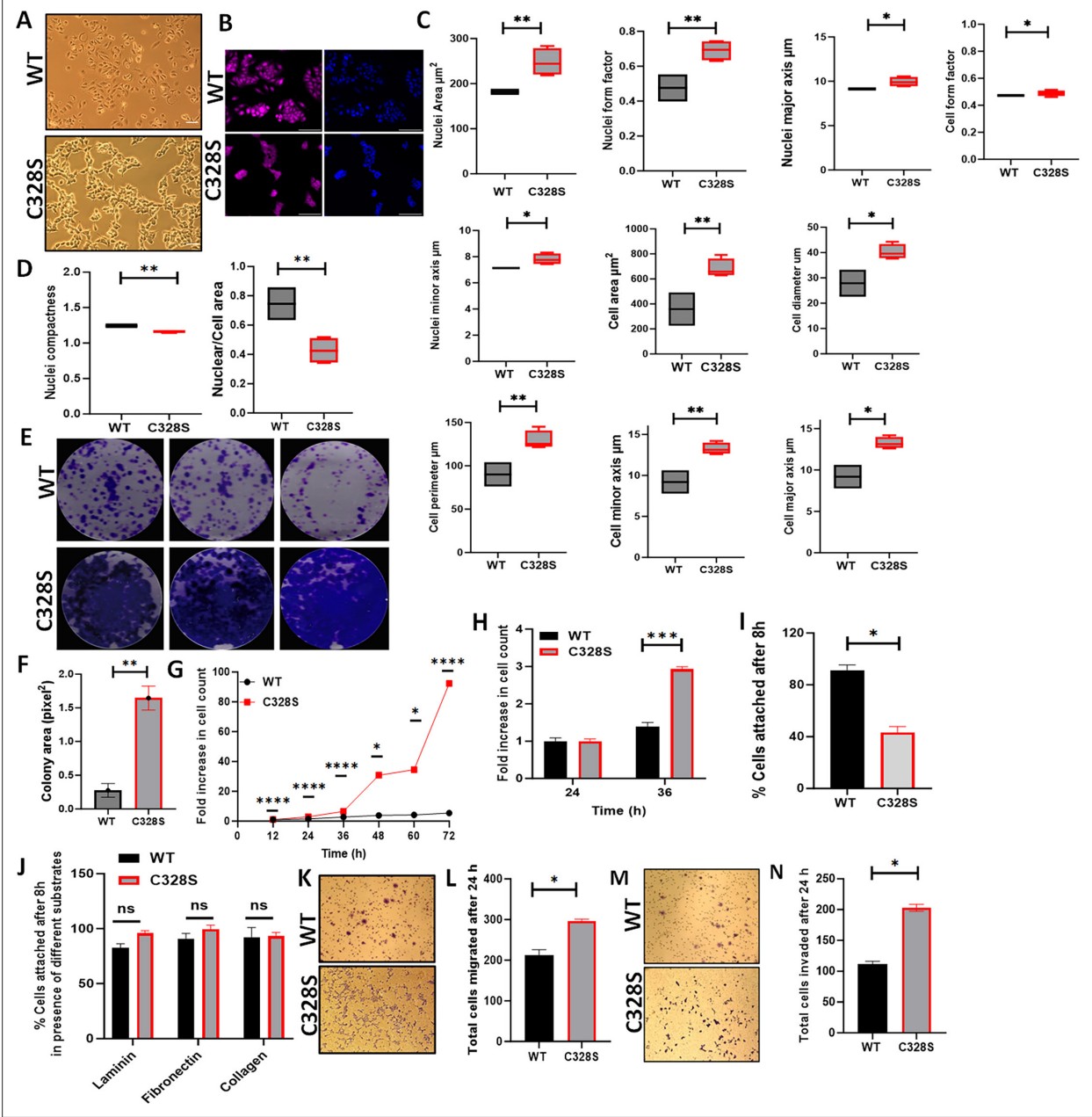

**Figure 2.** Effect of C328S-VIM on cell morphology, proliferation, adhesion, and invasion. (**A**) Morphology of MCF-7-expressing WT-VIM and C328S-VIM in brightfield (scale bar = 100 μm). (**B**) Morphology of WT and C328S cells stained with CellMask deep red dye. Images were captured by INCA 2200 and analysed using INCarta software (scale bar = 50 μm). (**C**) Differences in nuclear area, nuclei form factor, nuclear major axis, nuclear minor axis, cell area, cell diameter, cell perimeter, cell minor axis, cell major axis, and cell form factor between the two cell lines. (**D**) Significant reduction in nuclear compactness and nuclei/cell area between the two cell lines. Proliferation rate was compared between WT and C328S cells by (**E, F**) colony, (**G**) MTT, and (**H**) CyQUANT assays. (**I**) CyQUANT cell adhesion assay was performed to compare the cell adhesion between WT and C328S cells without substrate, and (**J**) with the addition of laminin, fibronectin, and collagen, separately. (**K**) Chemotactic migration of the WT and C328S cells through 8.0 μm culture inserts. The cells were fixed and stained with 0.1% (w/v) crystal violet before imaging. (**L**) The cells on the outer surface of the inserts were counted and compared between WT and C328S. (**M**) Chemotactic invasion in WT and C328S cells through 8.0 μm culture inserts coated with Matrigel. The cells were fixed and stained with 0.1% (w/v) crystal violet. (**N**) Total number of cells invaded on the outer surface of the inserts was counted. Statistical analyses: n = 3, error bars = ± SEM, Student's *t*-test was used to calculate p values using Microsoft Excel and are given by asterisks (*p<0.05, **p<0.01, ***p<0.001, and ****p<0.0001).

in C328S compared with the WT cells (*Figure 2K and L*). Similarly, the invasive capacity of the two cell lines was compared by coating the 8 μm pore size culture inserts with Matrigel. The results indicated a significantly higher number of cells invading through Matrigel in C328S-VIM (p<0.05) compared with the WT-VIM (*Figure 2M and N*).

## Upregulation of EMT and cancer stemness related signatures by C328S-VIM in MCF-7

As C328S-VIM significantly increased the proliferation of MCF-7, reduced their adhesion capacity, and increased migration and invasion towards FCS, we investigated the transcriptome profile of C328S-VIM vs WT-VIM-expressing MCF-7 cells using RNA-Seq in order to understand the underlying mechanism for the changes observed. The analysis showed that a total of 3421 out of 22,645 genes were significantly upregulated, whereas 3940 genes were downregulated (*Figure 3A*). Functional Gene Ontology (GO) analysis revealed that the significantly downregulated genes are involved in cell–cell adhesion, DNA packaging complex, and keratinocyte differentiation (*Figure 3B*). The most upregulated cellular function was pattern-specific processes, regionalisation, and related to development (*Figure 3C*). The most upregulated gene was *XIST* (*Figure 3D*)**,** a long non-coding oncogenic RNA (lncRNA) that is implicated in a large number of tumours (*Figure 3E*; *Madhi and Kim, 2019*). The RNA-Seq data was validated using RT-qPCR, and the regression analysis between two data sets showed significant Pearson correlation ($R^2$=0.77, p=0.003, Pearson *r*=0.88; *Figure 3F*). The upregulated and downregulated DEGs are provided in *Supplementary files 1 and 2*, respectively. *Supplementary files 3 and 4* enlist the upregulated and downregulated lncRNAs, respectively. The widespread downregulation of keratin gene expression shown by RNA-Seq and qPCR was corroborated by immunostaining of K8 and K18 (*Figure 3G*) and western blotting (*Figure 3H*). The quantification of the relative band intensity showed significant downregulation in K8, K18, K19 and upregulation of TWIST1 and CDH2/N-cadherin in C328S-VIM-expressing MCF-7 cells compared with the WT-VIM-expressing cells (*Figure 3I*).

The most upregulated and downregulated cellular functions in GO analyses are listed in *Figure 3—figure supplement 1*, *Figure 3—figure supplement 1—source data 1*and *Figure 3—figure supplement 1—source data 2*, *Figure 3—figure supplement 2*, and *Figure 3—figure supplement 2—source data 1* and *Figure 3—figure supplement 2—source data 2*, respectively. The KEGG pathway analysis is shown in *Figure 3—figure supplement 3* and *Figure 3—figure supplement 3—source data 1–3*. The RNA-Seq analysis shows that multiple epithelial markers were downregulated, and mesenchymal markers were upregulated among DEGs, indicating acquisition of EMT-like characteristics by C328S cells (*Figure 3—figure supplement 4*, *Figure 3—figure supplement 4—source data 1*and *Figure 3—figure supplement 4—source data 2*). Taken together, the data from RNA-Seq experiments begin to provide a mechanistic explanation for the structural and functional properties that we found to be associated with the C328S mutation in vimentin.

The RNA-Seq data showed upregulation of EMT transcription factors, including *ZEB1*, *ZEB2*, *TWIST 1*, *TWIST 2*, *ETS1*, *LEF1*, *SNAI2*, and *FOXC2*, although *SNAI1* was downregulated (*Figure 3J*). The upregulated differentially expressed breast cancer stem cell markers included *CD56/NCAM1*, *POU5F1*, *PROCR/CD201*, *ITGA6*, and the downregulated were *ESR1*, *PGR*, *HER2/ERBB2*, *ABCG2*, *CD44*, *CD24*, *EPCAM*, and *CDH1* (*Figure 3K*). Out of the presence or absence of three of them, *ESR1* (oestrogen receptor), *PGR* (progesterone receptor), and *HER2/ERBB2* (human epidermal growth factor receptor 2/HER2) make a breast cancer triple positive or triple negative. Our results show that MCF-7, which are triple-positive cells (*Comşa et al., 2015*), became triple reduced by the expression of C328S-VIM. Collectively these results imply that C328S-VIM in MCF-7 induces EMT-like characteristics and increases cancer stemness.

To confirm the expression of breast cancer stem cell markers identified as being differentially expressed in RNA-Seq analysis, FACS analysis was performed. Cells were stained for surface antigens with RY586–conjugated antibody specific for CD56/NCAM1, APC-conjugated antibody specific for CD201/PROCR, and the viability stain FVS575V was used for detecting the live cells. Gates were applied for cells (SSC-A:FSC-A), single cells (SSC-W: FSC-A), and live cells (FVS575V::L/d:FSC-A) for WT and C328S cells (*Figure 3—figure supplement 5A–F*). Data analysis using FlowJo v10 confirmed significant upregulation of breast cancer stem cell markers CD56/NCAM1 and CD201/PROCR in

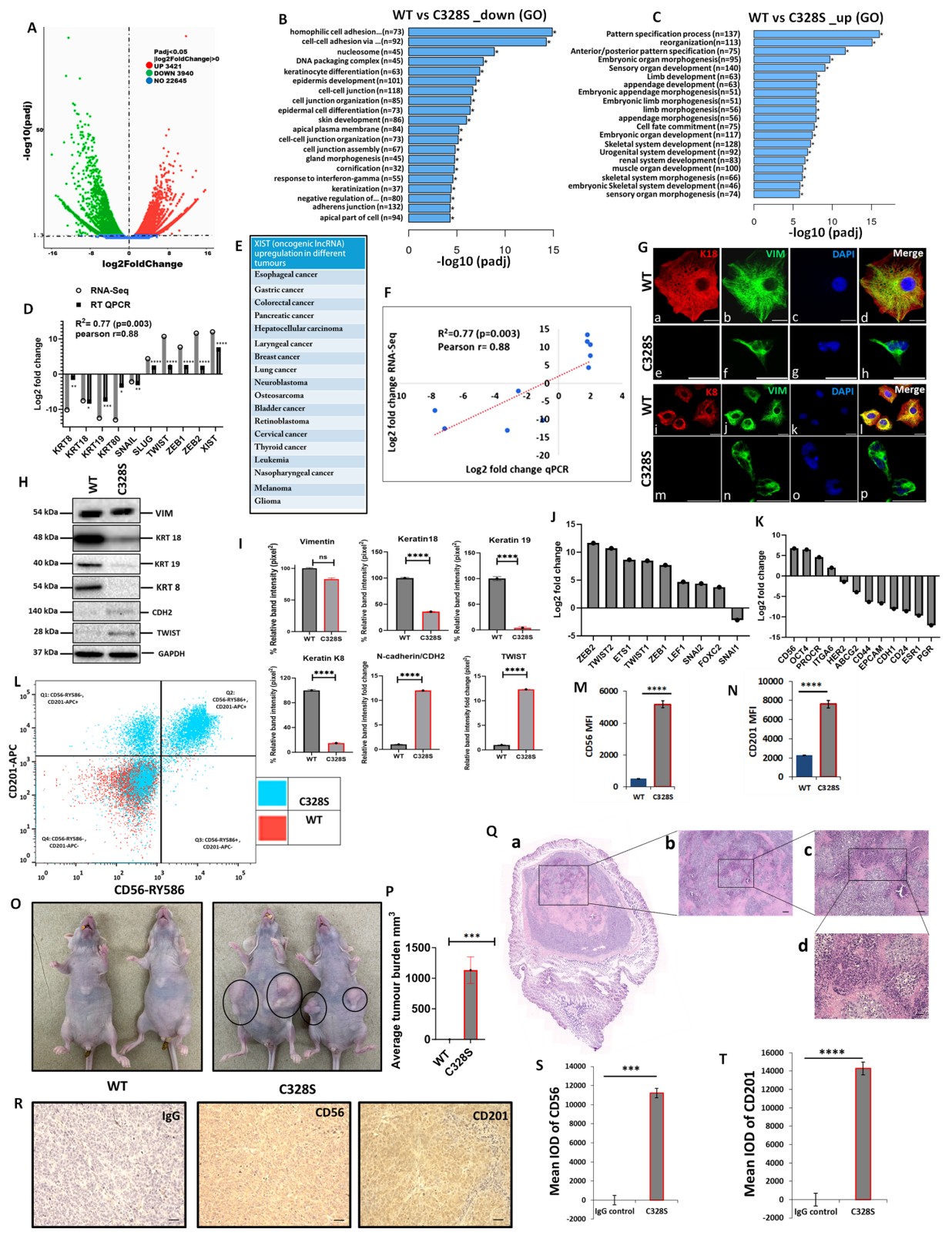

**Figure 3.** Transcriptomic insight into tumorigenic potential induced by C328S-VIM both in vitro and in vivo. (**A**) Volcano plot showing differentially expressed genes (DEGs) between WT and C328S cells. Gene Ontology (GO) showing overview of cellular functions, (**B**) downregulated, and (**C**) upregulated by DEGs. (**D**) RNA-Seq analysis showing log2-fold expression and validation by RT-qPCR for DEGs of interest. (**E**) *XIST*, the most upregulated gene, has been implicated in a large number of solid tumours (**Madhi and Kim, 2019**). (**F**) Linear regression analysis of log2-fold changes

*Figure 3 continued on next page*

*Figure 3 continued*

from RT-qPCR and RNA-Seq of DEGs. (**G**) Immunostaining of WT and C328S cells with V9, rabbit anti-K8, and rabbit anti-K18 using AF-488 (green) goat anti-mouse and AF-594 (red) goat anti-rabbit were used. Nuclei are in blue, overlapping images are shown as Merge. Leica DM4000B Epi-fluorescence microscope was used for imaging (scale bar = 20 μm). (**H**) VIM, K18, K19, K8, CDH2/N-cadherin, and TWIST1 expression by western blotting in WT and C328S cells. Relevant bands were cropped from the original blots shown in *Figure 3—source data 1* and *Figure 3—source data 2*. (**I**) Quantification of the protein expression in panel (**H**) using ImageJ. (**J**) Relative log2-fold changes in the expression of EMT transcription factors, and (**K**) breast cancer stem cell markers in WT and C328S cells by RNA-Seq analysis. (**L**) Flow cytometry overlay dot plot of CD56-RY586 vs CD201-APC after gating on single and live cells for immunophenotype. (**M**) Comparison of mean fluorescence intensity (MFI) of CD56 in WT and C328S cells. (**N**) Comparison of MFI of CD201 in WT and C328S cells. (**O**) Transplantation of WT and C328 cells in nude mice without oestrogen. (**P**) Average tumour burden after 2 weeks in nude mice injected with WT and C328S cells. (**Q**) H&E-stained tumour sections scale bar = 50 μm. (**R**) Representative image from immunohistochemical staining of CD56 and CD201 in tumour sections compared with IgG control, scale bar = 50 μm. (**S**) Quantification of CD56 and (**T**): CD201 staining in tumour sections and control using ImageJ. Statistical analyses: n = 3, Error bars = ± SEM, Student's *t*-test to calculate p values using Microsoft Excel and are given as asterisks (*p<0.05, **p<0.01, ***p<0.001, and ****p<0.0001).

The online version of this article includes the following source data and figure supplement(s) for figure 3:

**Source data 1.** Full-size western blots indicating the relevant bands cropped for *Figure 3H*.

**Source data 2.** Original files for western blots analysis displayed in *Figure 3H*.

**Figure supplement 1.** Gene Ontology (GO) pathway enrichment analyses of differentially expressed genes (DEGs) (upregulated) in WT vs C328S cells by RNA-Seq.

**Figure supplement 1—source data 1.** Spreadsheet showing all upregulated cellular functions by GO pathway enrichment analysis.

**Figure supplement 1—source data 2.** Spreadsheet showing significantly upregulated cellular functions by GO pathway enrichment analysis.

**Figure supplement 2.** Gene Ontology (GO) pathway enrichment analyses of differentially expressed genes (DEGs) (downregulated) in WT vs C328S cells by RNA-Seq.

**Figure supplement 2—source data 1.** Spreadsheet showing GO pathway enrichment analysis (all downregulated cellular functions by DEGs).

**Figure supplement 2—source data 2.** Spreadsheet showing GO pathway enrichment analysis (significantly downregulated cellular functions by DEGs).

**Figure supplement 3.** KEGG pathway analysis.

**Figure supplement 3—source data 1.** Spreadsheet showing downregulated cellular functions by KEGG analysis.

**Figure supplement 3—source data 2.** Spreadsheet showing upregulated cellular functions by KEGG analysis.

**Figure supplement 3—source data 3.** Spreadsheet showing all altered cellular functions by KEGG analysis.

**Figure supplement 4.** Differentially expressed genes (DEGs) deduced from the RNA-Seq data.

**Figure supplement 4—source data 1.** Spreadsheet containing upregulated genes.

**Figure supplement 4—source data 2.** Spreadsheet containing downregulated genes.

**Figure supplement 5.** Representative gating strategies for flow cytometry analysis of breast cancer stem cell markers CD56/NCAM1 and CD201/PROCR in WT and C328S cells.

**Figure supplement 5—source data 1.** Western blots indicating the relevant bands cropped for *Figure 3—figure supplement 5G*.

**Figure supplement 5—source data 2.** Original files for western blots analysis displayed in *Figure 3—figure supplement 5G*.

C328S-VIM-expressing MCF-7 cells. MFI of CD56 and CD201 in C328S-VIM cells was significantly higher (p<0.0001) compared with WT-VIM cells (*Figure 3L–N*).

Next, to examine tumorigenic potential in vivo, we injected WT and C328S cells subcutaneously and bilaterally into the flanks of 6-week-old female athymic nu/nu mice. C328S cells were able to produce tumours without oestrogen; however, WT cells did not produce any tumour (*Figure 3O and P*). These results confirm the tumorigenic potential of C328S-VIM that is oestrogen independent. Furthermore, to investigate breast cancer stem cell markers expression, routine H&E (*Figure 3Q*) and immunohistochemistry (*Figure 3R*) were performed on the tumour tissue sections for CD56 and CD201 expression, respectively. Normal mouse IgG was used as staining control. Tumour sections from C328S injected mice showed significantly higher IOD for CD56 (p<0.001) (*Figure 3S*) and CD201 (p<0.0001) (*Figure 3T*) compared with IgG control. These results confirmed our in vitro transcriptome and FACS analyses data.

To establish the specificity of tumour production by C328S-VIM, we created another single substitution, Y117L, at the beginning of the rod domain and a double substitution containing C328S and Y117L in the same vimentin molecule (referred to as DMT in this article). The expression of vimentin containing these substitutions is shown in *Figure 3—figure supplement 5—source data 1* and *Figure 3—figure supplement 5—source data 2*. Y117L substitution has been reported to assemble into

ULFs (*Robert et al., 2015*) and does not form long filaments. Interestingly transplantation of MCF-7Y117L-VIM did not induce tumour formation in nude mice, whereas MCF-7DMT-VIM did produce tumours (*Figure 3—figure supplement 5H*). These experiments suggest that tumour progression by C328S was specific and in DMT it was dominant over Y117L.

## shRNA-mediated downregulation of mutant vimentin or *XIS*T in C328S-VIM-expressing cells inhibits cancer potential

To determine whether the effects of C328S-VIM in MCF-7 can be reversed, we downregulated mutant vimentin or *XIST* in C328S-VIM cells by shRNA to more than 90 and 75%, respectively, as determined by RT-qPCR (p<0.01) (*Figure 4A and B*) and western blotting (p<0.001) (*Figure 4C and D*, *Figure 4—source data 1* and *Figure 4—source data 2*) compared with NTC. *XIST* RNA levels were reduced by 30% upon vimentin knockdown and *VIM* mRNA was downregulated by up to 20% (p<0.05) upon *XIST* knockdown (*Figure 4E and F*). CyQUANT adhesion assay showed that the adhesion capacity of cells treated with *VIM*-shRNA was significantly increased (p<0.001) (up to 54%±1.9) but not much with *XIST* shRNA compared to NTC in C328S-VIM cells (*Figure 4G and H*). Furthermore, cell proliferation (*Figure 4I and J*), migration (*Figure 4K and L*), and invasion (*Figure 4M and N*) were also decreased but the results were significant (p<0.05) only for the invasion assay. Morphological analysis of MCF-7C328S-shVIM cells showed that several features (not all, see *Figure 4—figure supplement 1*), including changes in cell area (p<0.01), nuclear/cell area (p<0.01), cell diameter (p<0.01), cell perimeter (p<0.01), cell major axis (p<0.05), and cell minor axis (p<0.05), were significantly reversed (*Figure 4O*). These results show that C328S-VIM is primarily responsible for the loss of adhesion and altered morphology of these cells. We also investigated the effect of mutant vimentin downregulation on breast cancer stem cell markers CD56/NCAM1 and CD201/PROCR in C328S-VIM-expressing cells upon shRNA treatment. Flow cytometry analysis of CD56 and CD201 expression upon shRNA treatment in these cells showed insignificant differences (*Figure 4P and Q*, *Figure 4—figure supplements 2 and 3*), demonstrating that the expression of cancer stem cell markers was irreversible.

## Discussion

Vimentin is a key player in several pathophysiological processes such as cell migration, proliferation, adhesion, stress response, EMT, and cancer metastasis (*Danielsson et al., 2018*). It has a single cysteine residue at position 328 that is considered a preferred site for post-translational modifications, and it is currently being intensely investigated due to its involvement in multiple cell functions, including filament assembly, cell polarity, elongation, and stress response to electrophiles and oxidants (*Viedma-Poyatos et al., 2020*). In spite of its likely importance in cellular physiology, the role of C328 in cancer progression and in determining the cancer stem cell phenotype has not been investigated before.

To investigate the significance of C328 in EMT and cancer, we made a mutant encoding a C328S substitution in vimentin (C328S-VIM) and transduced in MCF-7 cells, which are devoid of endogenous vimentin (*Usman et al., 2022a*; *Liu et al., 2015*; *Sivagurunathan et al., 2022*), and are widely used as a model to study EMT (*Guttila et al., 2012*; *Kondaveeti et al., 2015*). Structurally, the two amino acids, cysteine and serine, are similar except an electronegative sulphur atom (atomic radius 0.88 A°) in cysteine was replaced by another electronegative oxygen atom (atomic radius 0.48 A°) in serine. Although both atoms are electronegative, oxygen is slightly more electronegative (EN = 3.44) than sulphur (EN = 2.58). These seemingly minor differences were unlikely to cause major structural perturbations, so we were surprised when in silico simulations demonstrated that C328S substitution was likely to affect interactions between actin and vimentin. These computational predictions were confirmed by our immunofluorescence analyses of F-actin in C328S cells. Whereas WT vimentin-expressing cells showed normal stress fibres with no fragments or aggregates, in contrast C328S-VIM cells expressed aggregated and fragmented F-actin which was limited to the cortical margins of the cells with no stress fibre formation. These observations highlight the critical interactions of the rod domain of vimentin at C328 with actin that have never been reported, although earlier studies have demonstrated direct binding of the tail domain of vimentin with actin (*Esue et al., 2006*). It has recently been reported that C328 of vimentin modifies actin organisation in response to oxidants and electrophiles (*González-Jiménez et al., 2023*). Our data support these findings and suggest

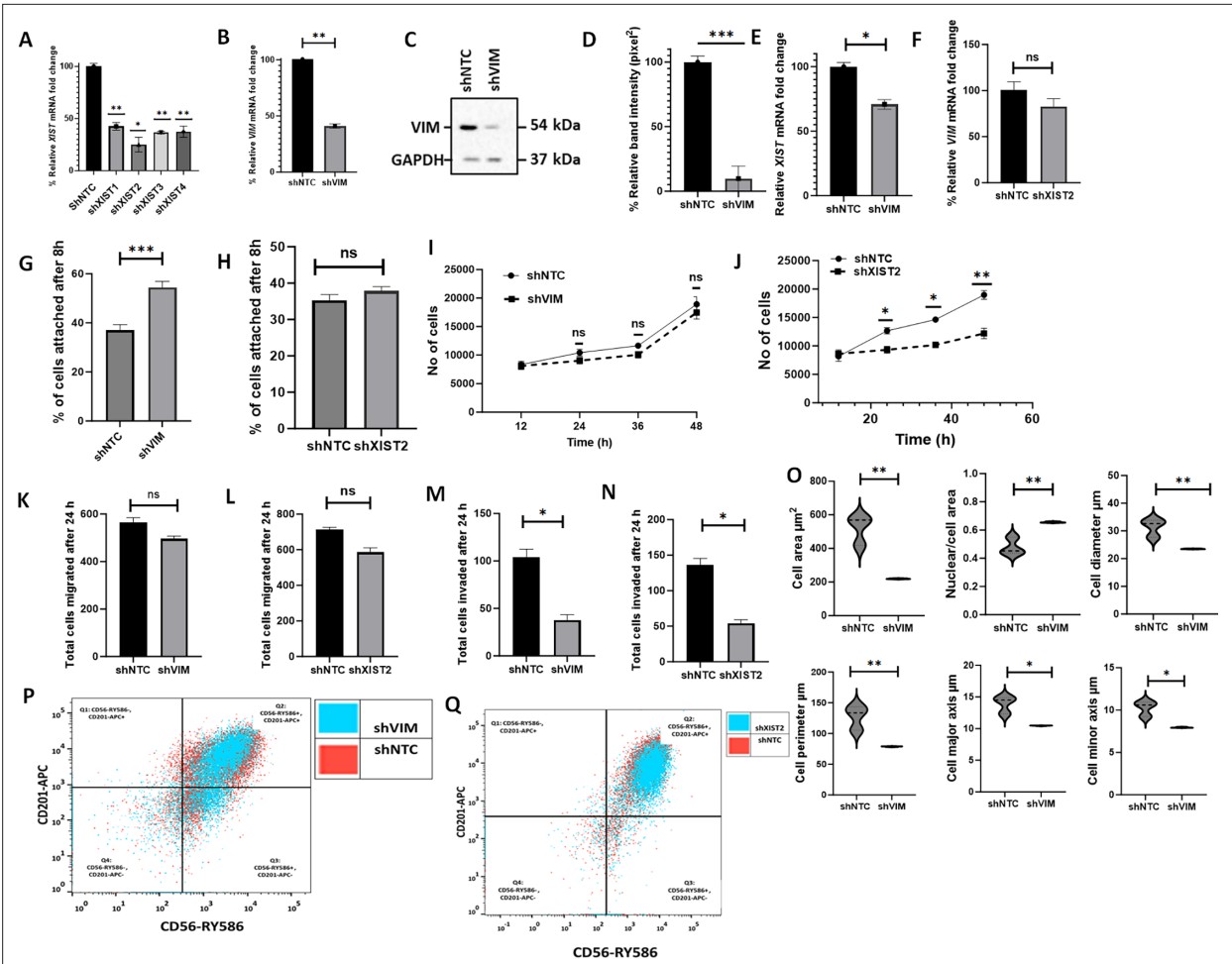

**Figure 4.** shRNA-mediated downregulation of *XIST* in C328S-VIM reverts cell phenotype. (**A**) Downregulation of *XIST* in C328S cells by four different shRNAs (*sh1-sh4*) for *XIST* or NTC by RT-qPCR. *shXIST2* was the most potent (p<0.05) compared with NTC. (**B**) VIM expression in MCF-7C328S_shVIM and MCF-7C328S_shNTC as determined by RT-qPCR. (**C**) Vimentin expression in MCF-7C328S_shVIM and MCF-7C328S_shNTC by western blotting (original blots in *Figure 4—source data 1* and *Figure 4—source data 2*). (**D**) Quantification of the protein expression in panel (**C**) using ImageJ. (**E**) *XIST* RNA in MCF-7 cells expressing C328S_shVIM and C328S_shNTC by RT-qPCR. (**F**) Relative *VIM* mRNA fold change (%) in MCF-7 cells expressing *C328S_shXIST2* and *C328S_shNTC* by RT-qPCR. Comparison of cell adhesion (**G**) between MCF-7C328S_shVIM and MCF-7C328S_shNTC, and (**H**) between MCF-7C328S_sh*XIST*2 and MCF-7C328S_shNTC cells without substrate by CyQUANT assay. Comparison of cell proliferation, (**I**) between MCF-7C328S_shVIM and MCF-7C328S_shNTC, and (**J**) between MCF-7C328S_sh*XIST*2 and MCF-7C328S_shNTC by MTT assay. Comparison of chemotactic migration (**K**) between MCF-7C328S_shVIM and MCF-7C328S_shNTC, and (**L**) between MCF-7C328S_shNTC cells through 8.0 μm culture inserts. Comparison of chemotactic invasion (**M**) between MCF-7C328S_shVIM and MCF-7C328S_shNTC, and (**N**) between MCF-7C328S_sh*XIST*2 and MCF-7C328S_shNTC cells through 8.0 μm Matrigel coated inserts. (**O**) Comparison of cell area, ratio of nuclei/cell area, cell diameter, cell perimeter, cell major axis, cell minor axis, between MCF-7C328S_shVIM and MCF-7C328S_shNTC. (**P**) Flow cytometry overlay dot plot of CD56-RY586 vs CD201-APC after gating on single and live cells for immunophenotyping of MCF-7C328S_shVIM and MCF-7C328S_shNTC cells. (**Q**) Flow cytometry overlay dot plot of CD56-RY586 vs CD201-APC after gating on single and live cells for immunophenotyping of MCF-7C328S_sh*XIST*2 and MCF-7C328S_shNTC cells. Statistical analyses: n = 3, error bars = ± SEM, Student's *t*-test was used to calculate p values using Microsoft Excel when two groups were compared, one-way ANOVA with Bonferroni test was applied using GraphPad Prism 10 when comparing more than two groups (*p<0.05, **p<0.01, and ***p<0.001).

The online version of this article includes the following source data and figure supplement(s) for figure 4:

**Source data 1.** Full-size western blots indicating the relevant bands cropped for *Figure 4C*.

**Source data 2.** Original files for western blots analysis displayed in *Figure 4C*.

**Figure supplement 1.** Morphology of C328SVIM_sh*XIST*2 and C328SVIM_shNTC cells.

**Figure supplement 2.** Flow cytometry analyses of C328S_shVIM and shNTC cells.

**Figure supplement 3.** Flow cytometry analyses of C328S_*shXIST*2 and *shNTC* cells.

**Figure supplement 4.** Significant impact of the C328S mutant vimentin on MCF-7 cells, leading to profound changes in their behaviour.

that a local perturbation at C328 (in this case by C328S substitution) can disrupt actin organisation even when cells are not exposed to oxidants and electrophiles. These observations have widespread implications, including cytoskeletal crosstalk, that regulates cell behaviour, especially during cancer development, progression, and spread.

Our data show that the C328S mutation in vimentin can change the overall morphology of MCF-7 cells (*Figure 2*). Earlier studies have implicated vimentin filaments in the maintenance of cell shape and increasing vimentin levels are positively related to a mesenchymal elongated phenotype and EMT (*Mendez et al., 2010*). In contrast, we have recently shown that the ectopic expression of the wild-type vimentin in MCF-7 can make them less elongated (*Usman et al., 2022a*). A possible explanation for these conflicting reports may be that cancer cells can express a wide spectrum of morphologies depending upon inducing factor and stage of EMT (*Leggett et al., 2016*; *Sinha et al., 2020*). As the C328S substitution in vimentin has increased the nuclear size and made the MCF-7 cells more rounded compared to WT vimentin, it is possible that the C328 residue may be an active player in vimentin-linked cell phenotypic changes since downregulation of the mutant vimentin significantly reversed the morphological changes (*Figure 4*).

We have previously shown that the expression of WT vimentin in MCF-7 induces cell migration without affecting proliferation or cell adhesion (*Usman et al., 2022a*). Here we show that C328S-VIM enhances cell proliferation, migration, invasion, and reduces cell adhesion, which are features closely associated with EMT-mediated cancer metastasis. Some of these changes could be reversed by RNAi-mediated downregulation of vimentin establishing a direct link between residue 328 of vimentin, or perhaps the surrounding region, in cancer progression, which is a novel finding. The increase in malignant characteristics of C328S cells implies that the original cysteine residue in vimentin acts as a tumour suppressor. In addition, our RNA-Seq, RT-qPCR, and western blot analyses showed upregulation of EMT-associated transcription factors (ZEB1, SNAI2, LEF1, ZEB2, TWIST1, TWIST2) and mesenchymal markers (CDH2, MMP2, FOXC2, FNDC1), as well as downregulation of epithelial markers (cytokeratins, CDH1, CLDN1, EPCAM), indicating possible EMT induction in C328S cells. There are three major keratins, including K8 (type II), K18, and K19 (both type I), that are expressed in MCF-7, and since all of them were downregulated (*Figure 3*), a central mechanism involved in the expression of all three keratins is likely to be affected by C328-VIM. One such mechanism is the involvement of transcription factor AP-1, which participates in the regulation of most keratin genes (*Blumenberg, 2000*). AP-1 is a complex of multiple different subunits (*Wu et al., 2021*), but our transcriptomic analysis suggests that none of these are affected by C328S-VIM (DEG lists provided in the supplementary material). It is therefore conceivable that C328S-VIM could suppress AP-1 activity, thereby switching off keratin gene expression. Further investigations are required to test this hypothesis.

Multiple breast cancer stem cell markers such as *POU5F1* (*Vezzoni and Parmiani, 2008*), *CD56* (*Moghbeli et al., 2014*), *CD49f* (*Zhang et al., 2020*), and *CD201/PROCR* (*Hwang-Verslues et al., 2009*) were also upregulated by C328-VIM, demonstrating increased stemness characteristics in MCF-7 cells as judged by the RNA-Seq analysis. Higher expression of the two stem cell markers, CD56 and CD201, was further validated by flow cytometry and also observed in vivo. RNA-Seq analysis also showed that MCF-7, which is a triple-positive (expression of ESR1, PGR, and HER2) cell line, became triple reduced (*Usman et al., 2021*) in the presence of the C328S-VIM. Breast oncologists broadly stratify breast cancers into triple positive, borderline, and triple negative (*Orrantia-Borunda et al., 2022*), which is indicative of their susceptibility to metastasise and their clinical course. Triple-positive breast cancers have a relatively good prognosis, followed by borderline and triple-negative lesions having by far the worst prognosis (*Orrantia-Borunda et al., 2022*). We have previously shown that wildtype vimentin does not affect the expression of these receptors in MCF-7 (*Usman et al., 2022a*). In the present study, however, C328S-VIM makes MCF-7, a triple-positive cell line (*Comşa et al., 2015*), into triple reduced, which has important clinical implications. It also suggests that a mechanism exists whereby disruption of molecular interactions between vimentin and actin caused by C328S, directly or indirectly, regulates the expression of these receptors, cancer progression, and metastasis.

The gene most upregulated by the C328S vimentin, as found in the RNA-Seq analysis, which is a long non-coding RNA *XIST,* is known to induce proliferation, invasion, and inhibit apoptosis in breast cancer (*Zong et al., 2020*). It is also reported to be upregulated in multiple solid tumours (*Madhi and Kim, 2019*; *Yang et al., 2021*). A recent study has unveiled its gatekeeping role in human mammary epithelium homeostasis, especially the differentiation aspect in human mammary stem cells (MaSCs)

(*Richart et al., 2022*). It therefore appears, based on our observations, that the vimentin C328S mutation is inducing EMT-like changes via *XIST* upregulation as downregulation of *XIST* in C328S cells significantly reduced proliferation and invasion.

A highly significant and interesting observation was that the presence of C328S-VIM in MCF-7, which are oestrogen-dependent tumorigenic cells (*Soule and McGrath, 1980*), made these cells oestrogen-independent in nude mice that further supports our in vitro data that C328S substitution had enhanced cancer stemness in MCF-7 cells. Expression of breast cancer stem cell markers CD56 and CD201 was evident in mouse tumours. These data suggest that the C328 in vimentin is an important regulator of cancer cell behaviour and that alterations in this region of the protein promote EMT-like changes and may induce metastasis (*Figure 4—figure supplement 4*).

Recently, wildtype vimentin has been shown to increase malignant progression in lung cancer. However, our data suggest that wildtype vimentin is non-tumorigenic in breast cancer, inferring that the effect of vimentin in cancer may be tissue specific, which has never been reported (*Berr et al., 2023*). Previous studies have described the importance of the C328 residue in filament assembly and organisation, stress response, aggresome formation, and lysosomal positioning only in SW13/cl.2 vimentin-deficient cells (*Pérez-Sala et al., 2015*; *Viedma-Poyatos et al., 2020*). However, the use of SW13/cl.2 cell model may not be appropriate to study the role of C328-VIM in EMT because these cells do not express any IFs. For a cancer cell to undergo EMT, it must be an epithelial cancer cell-expressing keratin. In that respect, the use of MCF-7 in this study is most appropriate because it does not express endogenous vimentin. As mutations in vimentin have been reported in patients around the area of C328 (*Usman et al., 2021*), our hypothesis that this mutation induces conformational changes that activate the complex processes of EMT in breast cancer cells would have clinical implications.

## Materials and methods

### Cell culture and cell lines

MCF-7, A431, and HFF cells were obtained from the cell bank of Cancer Research UK and characterised by the expression of key biomarkers using immunocytochemistry and reverse transcription (RT)/quantitative polymerase chain reaction (qPCR). They were cultured in Dulbecco's Modified Eagle Medium (DMEM), containing 10% (v/v) FCS, 50 units/mL penicillin and 50 μg/mL streptomycin (complete medium), and maintained in the incubator in an atmosphere of 5% $CO_2$ + 95% air at 37°C. Tissue culture cells were routinely tested for mycoplasma contamination using a commercially available kit.

### In silico analysis

Using PyMOL, an in silico analysis was conducted on wildtype and mutant vimentin to assess their interactions with wildtype actin. The PDB file for the wildtype vimentin (PdB id: P08670) was acquired from https://www.rcsb.org/, and the mutation (C328S) was introduced by subsequent energy minimisation steps.

### cDNA synthesis and qPCR

Cells were cultured in six-well plate format in triplicates and lysed in 500 μL Dynabeads mRNA lysis buffer when about 70% confluent. A total of 50 ng mRNA from each cell line was used to prepare the cDNA as described previously (*Usman et al., 2022a*). qPCR was performed with a LightCycler 480 qPCR System (Roche, Burgess Hill, UK) as described previously (*Usman et al., 2022a*). The forward and reverse primers for the genes studied are listed in *Supplementary file 5*.

### Protein extraction and western blotting

Cells were cultured in six-well plates in triplicates and lysed using 250 μL/well Laemmli lysis buffer, SDS gel electrophoresis, and western blotting were performed as described previously (*Usman et al., 2022a*). All antibodies used are listed in *Supplementary file 6*. Quantification of the band intensities was performed using ImageJ (*Rueden et al., 2017*).

### Plasmid constructs, cDNA cloning, retrovirus production, and spinfection

The full-length human vimentin cDNA was subcloned in pLPChygro as described previously (*Usman et al., 2022b*) and named as wildtype (WT) VIM. HPLC-purified primers (listed in *Supplementary*

file 7) were used for site-directed mutagenesis (SDM) at C328 and Y117 as described earlier (*Usman et al., 2024*). For preparing vimentin cDNA with double mutant (DMT), Y117L and C328S mutations were introduced sequentially and confirmed by sequencing.

For vimentin knockdown, cloning of *VIM*-shRNA along with non-target control (NTC) in *pSiren-Retro-Q* (Clontech, USA) has been described previously (*Paccione et al., 2008*). For silencing *XIST* expression, shRNA oligos were designed using the following software freely available online (http://web.stanford.edu/group/markkaylab/cgi-bin/). The forward and reverse primers (listed in *Supplementary file 7*) were annealed, phosphorylated, and ligated into *pSuper.retro.puro* previously digested with *Bgl*II and *Xho*I as described earlier (*Jamal et al., 2024*). All clones were sequenced before use. For packaging puromycin constructs, a one-step amphotropic retrovirus production was employed using Phoenix-A cells (*Swift et al., 2001*). For hygromycin constructs, a two-step method making ecotropic retrovirus using Phoenix-E in the first step followed by an amphotropic virus production using PT67 (*Miller and Chen, 1996*) in the second step was employed (*Aldehlawi et al., 2019*).

50,000 MCF-7 cells were seeded in T25 culture flask in complete medium, and retroviral transduction by spinfection was performed as described previously (*Usman et al., 2022a*). All transduced cell lines are listed in *Table 1*.

## Analysis of cell parameters

For analysing morphological features, 10,000 cells were seeded in 96-well plates in triplicates, and stained with CellMask Deep Red (Invitrogen, cat# H32721) (1 µL in 200 mL) and DAPI (1 µg/mL) for 2 h and washed with PBS 3× times as described earlier (*Usman et al., 2022a*). Cell morphology was analysed using an INCA 2200 IN Cell Analyzer GE Widefield System. More than 2000 cells were counted from each cell line using GE IN Carta software (INCarta Cytiva, USA).

## Functional assays

Colony formation assay, CyQUANT cell proliferation, CyQUANT cell adhesion, chemotactic migration, and MTT assays were performed as described earlier (*Usman et al., 2022a*; *Kumar et al., 2018*). For the invasion assay, culture inserts (8 µm pore size) were coated with Matrigel (1:20) in a 24-well plate in serum-free medium in triplicate, and the same procedure was followed as described for the migration assay (*Usman et al., 2022a*).

## RNA-Seq analysis and bioinformatics

Cells were plated in 10 cm dishes in duplicates, and, when they reached 80–90% confluence, they were washed twice with PBS and total RNA was extracted using a QIAGEN RNeasy Kit (cat# 74104) according to the manufacturer's protocol. Samples were processed by Novogene Europe (Cambridge, UK) for library preparation and bioinformatics analyses (*Usman et al., 2022a*).

## Transplantation assay

All transplantation experiments were carried out at Augusta University under Institutional Animal Care and Use Committee (IACUC) protocol number 2015-0736 in animal biosafety level 2 (ABSL2) conditions. Stably transduced MCF-7 cells (MCF-7CV, MCF-7WT, MCF-7C328S, MCF-7Y117L, and MCF-7DMT) were cultured under standard growth conditions as outlined earlier until they reached 80% confluence. Cells were trypsinised, counted, and resuspended in complete growth medium at a density of $1 \times 10^7$ cells per ml. Subsequently, cells were transplanted subcutaneously and bilaterally into the flanks of 6-week-old female athymic nu/nu mice ($2.5 \times 10^6$ cells/0.25 mL), two mice per cell line. Mice were monitored daily for general health and tumour formation. Once tumours became evident, caliper measurements were made weekly until the experimental endpoint. Tumour volume was determined for all groups using the formula: $V = (W^2 \times L)/2$, where V is the tumour volume, W is the tumour width, and L is the tumour length. Mice were euthanised by $CO_2$ inhalation followed by bilateral thoracotomy. Tumours were excised, fixed in formalin, paraffin-embedded, sectioned at 5 µm, stained with H&E, and imaged using a Keyence BZ-X700 microscope.

## Flow cytometry analyses

Cells were harvested by trypsinisation, centrifuged at $600 \times g$ for 5 min, washed 3× in PBS (supplemented with 0.5% BSA), and resuspended in appropriate volume of Flow Cytometry Staining Buffer

(FCSB) (eBioscience cat# 00-4222-57) to a final cell concentration of $1 \times 10^7$ cells/mL. Cells were stained for surface antigens CD56 and CD201 using antibodies listed in *Supplementary file 6* in FCSB so that the final volume was 100 µL (i.e., 50 µL of cell sample + 50 µL of antibody mix) for 30 min at 2–8°C (in the dark). The labelled cells were washed twice in FCSB and centrifuged at $600 \times g$ for 5 min at room temperature. The cell pellet was resuspended in azide-free and serum/protein-free PBS and 1 µL of Fixable Viability Dye (FVD) (listed in *Supplementary file 6*) was added per 1 mL of suspension. Cells were vortexed and incubated for 30 min at 2–8°C in the dark. These cells were washed twice with FCSB and suspended in 0.5 mL FCSB and analysed using a BD LSRII Analyzer in the Blizard Flow Cytometry Core Facility at QMUL. Data analyses were carried out using FlowJo v10.

## Immunohistochemistry

For the detection of stem cell markers in mouse tumours, 5 µm tissue sections were dewaxed in Safe-Clear (Fisher Scientific, Pittsburgh, PA), rehydrated, and antigen retrieval was performed by incubation in Retrievagen A (BD Biosciences, CA) at 95°C for 30 min. After cooling to ambient temperature, endogenous peroxidase was blocked by incubation in 3% (v/v) hydrogen peroxide for 15 min, slides were washed in Tris-buffered saline (TBS) pH 8.0, and blocked in normal goat serum (Vector Laboratories, Burlingame, CA). Sections were then incubated with anti-CD56 or anti-CD201 antibodies listed in *Supplementary file 6* or the equivalent concentration of normal rabbit or normal goat IgG as control, at 4°C for 16 h. Slides were washed in TBS, and then incubated sequentially with biotinylated anti-rabbit/mouse (for CD56) or biotinylated anti-goat (for CD201) secondary antibodies and streptavidin-peroxidase reagent (Vectastain Elite ABC-HRP Kit; Vector Laboratories, cat# PK-6100) according to the manufacturer's instructions. Colour development was achieved using 3, 3'-diaminobenzidine (DAB) substrate, and slides were lightly counterstained with Harris haematoxylin, dehydrated, mounted in Permount, and imaged by brightfield microscopy (Keyence Corporation, Itasca, IL). Routine H&E staining of tumour sections was performed by the Electron Microscopy and Histology Core Laboratory, Medical College of Georgia, Augusta University, as described previously (*Shahoumi et al., 2020*). The integrated optical density (IOD) of CD56 and CD201 was calculated as the product of optical intensity of positive cells × area of positive cells using ImageJ (*Crowe and Yue, 2019*).

## Statistical analyses

All experiments were performed in triplicates (technical repeats). To compare the two groups, two-tailed Student's *t*-tests were applied on raw data using Microsoft Excel, and p values <0.05 ($p<0.05$) were considered significant. To compare more than two groups, ordinary one-way or two-way analysis of variance was performed in combination with Bonferroni's test using GraphPad Prism 10. Linear regression analyses and Pearson correlation coefficients (Pearson's r) were determined using data analysis ToolPak in Microsoft Excel. All the results were represented as the mean of three individual experiments (n=3) with standard error of the mean (± S.E.M).

## Conclusion

In summary, this study highlights that vimentin is no longer a mere marker of EMT and metastasis, but appears to be an active participant in cancer progression. The strong correlation between the single cysteine residue at position 328 in vimentin with actin organisation, *XIST* induction, and hyperproliferation associated with breast oncogenesis is novel. This suggests that C328 in vimentin remodels actin cytoskeleton and protects against EMT and cancer growth via modulating lncRNA, *XIST*. Taken together, we propose that targeting vimentin via RNA interference should be considered a therapeutic strategy for breast cancer treatment.

## Acknowledgements

The authors are thankful to the Centre of Oral Immunobiology and Regenerative Medicine and the Blizard Institute for providing the research facilities necessary for this work. We are also thankful to Luke Gammon for help with the use of IN Cell Analyzer and Gary Warnes for FACS analysis. The authors also thank Professors Farida Fortune and Ian Mackenzie for useful discussion. We are also thankful to Mr Usman Baig for helping with figures. The authors would like to express their sincere gratitude to the Ministry of Health – Kuwait, represented by the Kuwait Embassy and the Kuwait Cultural Office in the United Kingdom, for their generous support. The publication of this research

paper was fully funded and sponsored through their commitment to advancing academic and scientific research.

## Additional information

### Funding

| Funder | Grant reference number | Author |
|---|---|---|
| Higher Education Commission, Pakistan | | Saima Usman |
| National Institute of Dental and Craniofacial Research | 5R01DE024381 | William Andrew Yeudall |

The funders had no role in study design, data collection and interpretation, or the decision to submit the work for publication.

### Author contributions

Saima Usman, Data curation, Formal analysis, Validation, Investigation, Visualization, Methodology, Writing – original draft; William Andrew Yeudall, Conceptualization, Investigation, Writing – review and editing; Muy-Teck Teh, Supervision, Writing – review and editing; Fatemah Ghloum, Investigation, Writing – review and editing; Hemanth Tummala, Data curation, Investigation, Writing – review and editing; Ahmad Waseem, Conceptualization, Supervision, Funding acquisition, Investigation, Visualization, Writing – original draft, Project administration, Writing – review and editing

### Author ORCIDs

Saima Usman https://orcid.org/0000-0002-2679-5628
William Andrew Yeudall https://orcid.org/0000-0002-5279-6333
Muy-Teck Teh https://orcid.org/0000-0002-7725-8355
Fatemah Ghloum https://orcid.org/0009-0007-5775-5518
Hemanth Tummala https://orcid.org/0000-0002-1413-745X
Ahmad Waseem https://orcid.org/0000-0002-7941-266X

### Ethics

All transplantation experiments were carried out at Augusta University under Institutional Animal Care and Use Committee (IACUC) protocol number 2015-0736, in animal biosafety level 2 (ABSL2) conditions.

Reviewer #2 (Public review): https://doi.org/10.7554/eLife.104191.3.sa1
Author response https://doi.org/10.7554/eLife.104191.3.sa2

## Additional files

### Supplementary files

Supplementary file 1. List of upregulated genes (cut off padj =0.00009).

Supplementary file 2. List of downregulated genes (cut off padj =0.00009).

Supplementary file 3. List of upregulated lnRNAs (cut off padj =0.00009).

Supplementary file 4. List of downregulated lnRNAs (cut off padj =0.00009).

Supplementary file 5. List of primers used for qPCR.

Supplementary file 6. List of primary and secondary antibodies used in this research work.

Supplementary file 7. List of primers used for making *XIST* shRNA constructs and site-directed mutagenesis at C328 and Y117.

MDAR checklist

## Data availability

All data generated or analysed during this study are included in the manuscript and supporting files; source data files have been provided in supplementary files.

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
