## [Editor Report · eLife Assessment]

This **valuable** study reveals that the structural protein vimentin promotes the epithelial–mesenchymal transition in breast cancer cells. Utilising robust and validated methodologies, the data collected provide a **solid** foundation for further investigation into metastasis models. This work will be of significant interest to researchers in the field of breast cancer.

---

## [Referee Report · Reviewer #2 (Public review)]

The aim of the investigation was to find out more about the mechanism(s) by which the structural protein vimentin can facilitate the epithelial-mesenchymal transition in breast cancer cells.

The authors focused on a key amino acid of vimentin, C238, its role in the interaction between vimentin and actin microfilaments, and the downstream molecular and cellular consequences. They model the binding between vimentin and actin in silico to demonstrate the potential involvement of C238, due to its location in a rod domain known to bind beta-actin. The phenotype of a non-metastatic breast cancer cell line MCF7, which doesn't express vimentin, could be changed to a metastatic phenotype when mutant C238S vimentin, but not wild-type vimentin, was expressed in the cells. Expression of vimentin was confirmed at the level of mRNA, protein and microscopically. Patterns of expression of vimentin and actin reflected the distinct morphology of the two cell lines. Phenotypic changes were assessed through assay of cell adhesion, proliferation, migration and morphology and were consistent with greater metastatic potential in the C238S MCF7 cells. Changes in the transcriptome of MCF7 cells expressing wild-type and C238S vimentins were compared and expression of Xist long ncRNA was found to be the transcript most markedly increased in the metastatic cells expressing C238S vimentin. Moreover changes in expression of many other genes in the C238S cells are consistent with an epithelial mesenchymal transition. Tumourigenic potential of MCF7 cells carrying C238S but not wild-type, vimentin was confirmed by inoculation of cells into nude mice. This assay is a measure of stem-cell quality of the cells and not a measure of metastasis. It does demonstrate phenotypic changes that could be linked to metastasis.

shRNA was used to down-regulate vimentin or Xist in the MCF7 C238S cells. The description of the data is limited in parts and data sets require careful scrutiny to understand the full picture. Down-regulation of vimentin reversed the morphological changes to some degree, but down-regulation of Xist didn't. Conversely down-regulation of Xist inhibited cell growth, a sign of reversing metastatic potential, but down-regulation of vimentin had no effect on growth. Down-regulation of either did inhibit cell migration, another sign of metastatic reversal. Most of these findings are consistent with previous work based on ectopic expression of wild-type vimentin in MCF7 cells, but the mechanism of inhibition of cell migration by downregulation of Xist remains speculative. More complete knockdown of vimentin or Xist by CRISPR technology may be helpful.

Overall the study describes an intriguing model of metastasis that is worthy of further investigation, especially at the molecular level to unravel the connection between vimentin and metastasis. The identification of a potential role for Xist in metastasis, beyond its normal role in female cells to inactivate one of the X chromosomes, corroborates the work of others demonstrating increased levels in a variety of tumours in women and even in some tumours in men. It would be of great interest to see where in metastatic cells Xist is expressed and what it binds to.

Comments on revisions:

The revised manuscript incorporates changes in presentation of the data modelling interaction between the region of vimentin including C238 and F-actin. There is also inclusion of an extra citation supporting the role for Xist in cancer stem cell differentiation.

---

## [Author Response]

The following is the authors’ response to the original reviews

**Reviewer #1 (Public review):**
Summary, and Strengths:The authors and their team have investigated the role of Vimentin Cysteine 328 in epithelial-mesenchymal transition (EMT) and tumorigenesis. Vimentin is a type III intermediate filament, and cysteine 328 is a crucial site for interactions between vimentin and actin. These interactions can significantly influence cell movement, proliferation, and invasion. The team has specifically examined how Vimentin Cysteine 328 affects cancer cell proliferation, the acquisition of stemness markers, and the upregulation of the non-coding RNA XIST. Additionally, functional assays were conducted using both wild-type (WT) and Vimentin Cysteine 328 mutant cells to demonstrate their effects on invasion, EMT, and cancer progression. Overall, the data supports the essential role of Vimentin Cysteine 328 in regulating EMT, cancer stemness, and tumor progression. Overall, the data and its interpretation are on point and support the hypothesis. I believe the manuscript has great potential.

The authors are thankful to the reviewers for carefully reading the manuscript and evaluating the data to make positive comments and supporting our conclusions.

Weaknesses:Minor issues are related to the visibility and data representation in Figures 2E and 3 A-F

We have revised the figures (Figure 2E and Figure 3A-F) to increase the data visibility.

**Reviewer #2 (Public review):**
The aim of the investigation was to find out more about the mechanism(s) by which the structural protein vimentin can facilitate the epithelial-mesenchymal transition in breast cancer cells.The authors focussed on a key amino acid of vimentin, C238, its role in the interaction between vimentin and actin microfilaments, and the downstream molecular and cellular consequences. They model the binding between vimentin and actin in silico to demonstrate the potential involvement of C238, but the outcome is described vaguely.

We have expanded the discussion of these results in the manuscript to more explicitly describe the critical role of C238 in the vimentin-actin interaction. Specifically, we highlight that C238 lies within a region of the vimentin rod domain known to mediate key protein-protein interactions. Our modeling shows that the thiol group of C238 enables specific hydrogen bonding and potential disulfide-mediated interactions with actin, which are disrupted upon mutation to serine. These findings provide mechanistic insight into the functional importance of this residue.

The phenotype of a non-metastatic breast cancer cell line MCF7, which doesn't express vimentin, could be changed to a metastatic phenotype when mutant C238S vimentin, but not wild-type vimentin, was expressed in the cells. Expression of vimentin was confirmed at the level of mRNA, protein, and microscopically. Patterns of expression of vimentin and actin reflected the distinct morphology of the two cell lines. Phenotypic changes were assessed through assay of cell adhesion, proliferation, migration, and morphology and were consistent with greater metastatic potential in the C238S MCF7 cells. Changes in the transcriptome of MCF7 cells expressing wild-type and C238S vimentins were compared and expression of Xist long ncRNA was found to be the transcript most markedly increased in the metastatic cells expressing C238S vimentin. Moreover changes in expression of many other genes in the C238S cells are consistent with an epithelial mesenchymal transition. Tumourigenic potential of MCF7 cells carrying C238S but not wild-type, vimentin was confirmed by inoculation of cells into nude mice. This assay is a measure of the stem-cell quality of the cells and not a measure of metastasis. It does demonstrate phenotypic changes that could be linked to metastasis.shRNA was used to down-regulate vimentin or Xist in the MCF7 C238S cells. The description of the data is limited in parts and data sets require careful scrutiny to understand the full picture. Down-regulation of vimentin reversed the morphological changes to some degree, but down-regulation of Xist didn't.

This is understandable given the fact that vimentin interacts with actin which is known to determine cell shape. *XIST* being a non-coding RNA will not have the same effect.

Conversely, down-regulation of *XIST* inhibited cell growth, a sign of reversing metastatic potential, but down-regulation of vimentin had no effect on growth.

*XIST* is known to get induced in a number of cancers (see Figure 3E) which is consistent with our observation that its downregulation will inhibit cell growth. However, downregulation of vimentin had no effect on growth which is consistent with our previously published observation that ectopic expression of wildtype vimentin in MCF-7 cells did not influence cell growth (Usman et al Cells 2022, 11(24), 4035; https://doi.org/10.3390/cells11244035).

Down-regulation of either did inhibit cell migration, another sign of metastatic reversal.

We have previously shown that ectopic expression of wildtype vimentin in MCF-7 stimulate cell migration due to downregulation of CDH5 (endothelial cadherins) (Usman et al Cells 2022, 11(24), 4035). Therefore, downregulation of vimentin is expected to inhibit cell migration which is what we observed in this study. Why downregulation of *XIST* inhibited cell migration is not clear. It is conceivable that *XIST* downregulation affects Lamin expression which may suppress intercellular interactions to increase cell migration. This hypothesis is supported by the fact that vimentin expression in MCF-7 affects Lamin expression (Usman et al Cells 2022, 11(24), 4035).

The interpretation of this type of experiment is handicapped when full reversal of expression is not achieved, as was the case in this study.

Full reversal of any biological effect is almost impossible to achieve which is because the shRNAs by nature are not 100% effective. This can however be tested using crispr Cas 9 gene editing to completely knockdown a protein (can’t be used for *XIST* as it is a non-coding RNA). In that case one has to assume that it will have no off-target effect.

Overall the study describes an intriguing model of metastasis that is worthy of further investigation, especially at the molecular level to unravel the connection between vimentin and metastasis. The identification of a potential role for Xist in metastasis, beyond its normal role in female cells to inactivate one of the X chromosomes, corroborates the work of others demonstrating increased levels in a variety of tumours in women and even in some tumours in men. It would be of great interest to see where in metastatic cells Xist is expressed and what it binds to.

The authors fully agree that it is an interesting model of metastasis/oncogenesis that requires further investigation.